# High Throughput Identification of Novel Conotoxins from the Vermivorous Oak Cone Snail (*Conus quercinus*) by Transcriptome Sequencing

**DOI:** 10.3390/ijms19123901

**Published:** 2018-12-05

**Authors:** Bingmiao Gao, Chao Peng, Yabing Zhu, Yuhui Sun, Tian Zhao, Yu Huang, Qiong Shi

**Affiliations:** 1Hainan Provincial Key Laboratory of Research and Development of Herbs, College of Pharmacy, Hainan Medical University, Haikou 571199, China; gaobingmiao@hainmc.edu.cn; 2Shenzhen Key Lab of Marine Genomics, Guangdong Provincial Key Lab of Molecular Breeding in Marine Economic Animals, BGI Academy of Marine Sciences, BGI Marine, BGI, Shenzhen 518083, China; pengchao@genomics.cn; 3Institute for Molecular Bioscience, The University of Queensland, St. Lucia, Brisbane, QLD 4072, Australia; 4BGI Genomics, BGI-Shenzhen, Shenzhen 518083, China; zhuyabing@genomics.cn (Y.Z.); sunyuhui@genomics.cn (Y.S.); 5Children’s Hospital of Philadelphia, Philadelphia, PA 19104, USA; 6Chemistry Department, College of Art and Science, Boston University, Boston, MA 02215, USA; domzhao@bu.edu; 7BGI Education Center, University of Chinese Academy of Sciences, Shenzhen 518083, China

**Keywords:** cone snail, conotoxin, *Conus quercinus*, transcriptome, venom duct, venom bulb, salivary gland

## Abstract

The primary objective of this study was to realize the large-scale discovery of conotoxin sequences from different organs (including the venom duct, venom bulb and salivary gland) of the vermivorous Oak cone snail, *Conus quercinus*. Using high-throughput transcriptome sequencing, we identified 133 putative conotoxins that belong to 34 known superfamilies, of which nine were previously reported while the remaining 124 were novel conotoxins, with 17 in new and unassigned conotoxin groups. A-, O_1_-, M-, and I_2_- superfamilies were the most abundant, and the cysteine frameworks XIII and VIII were observed for the first time in the A- and I_2_-superfamilies. The transcriptome data from the venom duct, venom bulb and salivary gland showed considerable inter-organizational variations. Each organ had many exclusive conotoxins, and only seven of all the inferred mature peptides were common in the three organs. As expected, most of the identified conotoxins were synthesized in the venom duct at relatively high levels; however, a number of conotoxins were also identified in the venom bulb and the salivary gland with very low transcription levels. Therefore, various organs have different conotoxins with high diversity, suggesting greater contributions from several organs to the high-throughput discovery of new conotoxins for future drug development.

## 1. Introduction

Cone snails (*Conus* spp.) are venomous predators living in tropical marine waters all over the world. They rapidly immobilize their prey using a complex cocktail of small and disulfide-rich peptides, usually 7–46 amino acids (aa) in length and collectively known as conotoxins or conopeptides [1,2]. Each cone snail is believed to synthesize 50–200 such peptides, and it is estimated that more than 80,000 different conotoxins may exist in approximately 800 species of cone snails around the world [2,3,4]. Most of these conotoxins can selectively target certain voltage-gated ion channels, ligand-gated ion channels, G-protein coupled receptors and neurotransmitter transporters in the central and peripheral nervous systems [5,6,7]. Thus, these conotoxins are widely used as pharmacological tools in neuroscience research, and some have already demonstrated therapeutic potential in preclinical and clinical trials [7]. The most famous, ω-MVIIA (Ziconotide) is derived from the venom of a fish-hunting species *C. magus*, and has been approved by the American Food and Drug Administration (FDA)for treatment of chronic pain in patients with cancer or acquired immune deficiency syndrome (AIDS) [8,9].

Although over 80,000 different conotoxins may exist, to date less than 0.1 percent of them have been characterized from 100 *Conus* species by traditional approaches [3,10,11]. However, these methods, which are used to isolate and identify the potential bioactive substances directly from venoms are now widely considered to be time-consuming, as having low sensitivity and to be limited by sample availability. In contrast, high-throughput sequencing can achieve higher sequencing depth (amplification with more sequencing reads) and greater coverage of transcriptomes, so that even rare transcripts with low transcription levels can be identified [12]. Therefore, several recent studies using next-generation sequencing technologies on the venom duct transcriptomes of *Conus* species have uncovered about 3000 conotoxin genes in 14 cone snails [1,2,13,14,15,16,17,18,19,20,21,22,23,24,25]. Altogether, these transcriptome studies show that each *Conus* species produces at least 100–400 different conotoxins, which has been very beneficial for the discovery of novel superfamilies and new cysteine frameworks.

To date, only 41 conotoxins have been previously identified in the Oak cone snail (*C. quercinus*) using traditional methods, and they were classified into 6 superfamilies (A, M, T, Q, O_1_ and K) [26,27,28]. To further contribute to the cataloguing of conotoxin diversity in the main lineages of cone snails, we employed the high-throughput transcriptome sequencing approach to explore additional conotoxins from the transcriptomes of different organs, including the venom duct, venom bulb and salivary gland in *C. quercinus*. For simplification, the three transcriptomes from the different organs were named as “VB” (venom bulb), “VD” (venom duct), and “SG” (salivary gland) datasets.

## 2. Results

### 2.1. Assembly of Transcriptome Sequences

Our high-throughput sequencing generated approximately 29.45, 30.18 and 30.78 million paired-end short reads with a length of 150 bp for VD, SG and VB, respectively. After removal of low-quality reads and adapter sequences, we obtained 4.04, 4.01 and 4.10 gigabases (Gb) of clean data for each sample, and the clean data ratio was estimated to be 91.48%, 88.63% and 88.83%, respectively (Table 1). De novo assembling of these clean reads produced 171 k, 225 k and 124 k contigs for VD, SG and VB, respectively, and N50 of each assembly was calculated to be 586, 669 and 651 bp in size. After clustering contigs into consensus unigenes based on sequence similarity, the final assembly of each transcriptome contained 91 k, 113 k and 66 k unigenes for VD, SG and VB, respectively, with similar GC contents (44.21%, 44.87% and 44.72%; Table 1, Appendix A).

To investigate the transcription levels of genes including conotoxins in each sample, we mapped the clean reads back to the de novo assembled unigenes (Appendix A) and estimated the abundance of each gene. There were 72 k, 85 k and 61 k transcribed genes in VD, SG and VB, respectively, (Appendix A) in which 33.49% (24,327 genes), 21.97% (18,801 genes) and 50.06% (30,676 genes; Appendix A) had high transcription levels (values of fragments per kilobase of exon per million fragments mapped (FPKM)≥ 10; see more details in Section 2.4).

Subsequently, functional annotations of the assembled transcripts were realized by blasting against several public databases and a total of 30,830 (23.11%) (Appendix A) were successfully annotated in at least one database, including 8136 (6.10%) genes with common hits in all databases (Appendix A). There were 11,719 genes enriched into 44 KEGG (Kyoto Encyclopedia of Genes and Genomes) pathways, with signal transduction being the most enriched one (Appendix A). Another 11,323 genes were assigned into 57 GO (Gene Ontology) terms, which can be classified into three categories, including biological process, cellular component and molecular function (Appendix A).

### 2.2. Summary of Conotoxins in the Three Transcriptomes

The putative conotoxin sequences were predicted by BLASTX search and HMMER analysis [24] against a local reference database of known conotoxins from the ConoServer database [29], and then checked manually using the ConoPrec tool [30]. After removal of the transcripts with duplication or truncated mature region sequences, we obtained a total of 65, 55 and 52 putative conotoxin transcripts for the three datasets of VD, SG and VB, respectively (Appendix A). We then combined the three conotoxin datasets into the total transcript dataset of *C. quercinus* (Qc-transcript) and named the predicted 133 conotoxins as Qc-001 to Qc-133 (Appendix A). They have at least one aa difference between each other in the mature regions. The 41 conotoxins previously identified by traditional methods in *C. quercinus* [26,27,28] were classified into six superfamilies, including A, M, T, Q, O_1_ and K (Figure 1A; Appendix A); 14 of these conotoxins belong to the A-superfamily, six belong to the M-superfamily and four belong to the T-superfamily, which account for 33.33%, 14.29% and 9.52%, respectively.

In the present study, each putative conotoxin was assigned to a superfamily based on percentage of sequence identity to the highly conserved signal regions of the known superfamilies in the ConoServer database (see more details in Section 4. Here, 133 novel putative conotoxins in *C. quercinus* were assigned to 34 known conotoxin superfamilies (Figure 1, Appendix A), in which 17 were described in the ConoServer database and the rest were identified by serial BLAST homology searches. In addition, seven belonged to a new superfamily (NSF) and 11 were unassigned to any superfamily. However, 68 conotoxins belonged to the majority of superfamilies (A, B_2_, C, E, J, I_2_, O_1_, O_2_, O_3_, L, M, P, T, W, Y and Z; Figure 1C), and 65 were classified into other families such as conkunitzin, conantokin, conoporin and con-ikot-ikot (Figure 1D). Interestingly, we discovered 12 new superfamilies (B_2_, C, E, I_2_, J, O_2_, O_3_, L, P, W, Y and Z) from the Qc-transcript, which have never been reported in this species before, although neither the Q- or K-superfamily was discovered. Similar to the ConoServer database, A-, M-, O-, and I_2_-superfamilies accounted for the major proportion of venom conotoxins in *C. quercinus* (Figure 1C).

The cysteine patterns of conotoxins in *C. quercinus* are summarized in Figure 1B,E. The 41 conotoxins previously identified by traditional methods have six types of cysteine frameworks, including I (CC-C-C), III (CC-C-C-CC), V (CC-CC), VI/VII (C-C-CC-C-C), XVI (C-C-CC) and XXIII (C-C-C-CC-C). In addition, other frameworks such as C-C, CC-C and C-C-C, were also identified. In contrast, except for the cysteine frameworks determined in the 41 conotoxins, nine other types of cysteine frameworks were also discovered from the other 64 conotoxins, including IX (C-C-C-C-C-C), VIII (C-C-C-CC-C-C-C-C-C), XI (C-C-CC-CC-C-C), XII (C-C-C-C-CC-C-C), XIII (C-C-C-CC-C-C-C), XIV (C-C-C-C), XVII (C-C-CC-C-CC-C), XXII (C-C-C-C-C-C-C-C) and XXVII (C-CC-C-C-C), respectively. Meanwhile, other limited frameworks, C-C, CC-C and C-C-C, were determined as well.

### 2.3. Comparison of Conotoxins in the Three Transcriptomes

The comparative distribution of putative conotoxins is summarized in Figure 2A. Of these putative conotoxins, seven were identified in all the three organs (VD, SG and VB), 13 were common to both VD and VB (but not in SG), 8 were shared by VB and SG (not in VD), and 5 were common to both VD and SG (not in VB). Among the 41 previously published conotoxin sequences of *C. quercinus* (Qc-known; Figure 2B and Appendix A), seven (Qc1.1b/Qc-072, Qc1.2/Qc-073, Qc3.1/Qc-077, Qc3-IP01/-Qc-100, Qc1.16/Qc-104, Qc5.3/Qc-105, and Qc5.2/Qc-118) were recovered in our present transcript dataset of *C. quercinus* (Qc-transcript; Figure 2B and Appendix A). In addition, only two (Qc-056 and Qc-093) of the 133 identified conotoxins have the same mature region sequences as previously reported in other *Conus* species (*C. lividus* and *C. caracteristicus*; other-known in Figure 2B), and they are both classified in the M-superfamily (Lv3-IP02 and Qc3-HGS02, respectively) [31].

Figure 2C shows the differences in spatial distribution of different conotoxin superfamilies among the VD, VB, and SG datasets of *C. quercinus.* Conotoxin sequences of A-, B_2_-, E-, I_2_, -M-, O_1_-, O_2_-, O_3_-, T-, Y- and Z-superfamilies were identified in VD, VB and SG, and A-, O_1_-, M- and I_2_-superfamilies were the most predominant conotoxin groups in *C. quercinus* (Figure 1 and Figure 2C). In addition, the J-superfamily was identified only in SG, not in VD and VB; C-, P- and M-superfamilies were not found in SG, and C-, J-, L-, and W-superfamilies were not found in VB (Figure 2C). Therefore, the venom duct, venom bulb and salivary gland had different conotoxins with high diversity. However, our results confirm that the venom duct is the most high-throughput factory for organizational manufacture of conotoxins.

### 2.4. Differential Transcription of Conotoxins in Different Organs

FPKM values were calculated to represent the transcription levels of conotoxins. We discovered that different organs have differential transcription profiles of toxins. The top 20 putative conotoxins (with the highest FPKM values) were selected from each of the three datasets for comparison. We observed that transcription levels in the VD (RPKM values above 961.5) were generally higher than those in the SG and VB (Figure 3A, Appendix A). In contrast, the highest RPKM values in the SG-top20 and VB-top20 datasets were 2364.27 and 247.12, respectively.

Among the top 20 conotoxins of the three datasets, six putative conotoxins were found in the lists of the VD-top20 and VB-top20, but not in the SG-top20 list (Figure 3B). Interestingly, these six putative conotoxins with high transcription levels belonged to different families with various cysteine patterns (Figure 3C). For example, Qc-108 with a framework of C-C belongs to an unassigned superfamily, although it was similar to the new Superfamily-4 conotoxin G127-VD from *C. geographus* [32] through sequence alignment (the identity is about 85%). Qc-093 with the framework III (CC-C-C-CC) belongs to the M-superfamily, identical to the MMSK group conotoxin Qc3-HGS02 from *C. caracteristicus* [31], and sharing 93% identity with the M-superfamily MMSK group conotoxin Qc3-YGS01 from *C. quercinus* [31]. Qc-126 with a framework of VI/VII (C-C-CC-C-C) belongs to the O_3_-superfamily, showing 81% identity with the O_3_-superfamily conotoxin Tr7.4 from *C. terebra*. This is the only O_3_-superfamily conotoxin identified in *C. quercinus*. Qc-114 with non-cysteine belongs to the conantokin superfamily, sharing 75% identity with the conantokin-F Fla-7 from *C. flavidus*. Qc-129 with the framework XVII (C-C-CC-C-CC-C) belongs to the Y-superfamily, sharing 70% identity with a conotoxin from *C. betulinus*. Qc-103 with the framework VI/VII(C-C-CC-C-C) belongs to Divergent_ M-L-LTVA, which showed 83% identity with a conotoxin from *C. betulinus* [24].

As expected, it appears that most of the identified conotoxins were synthesized at high levels in the venom duct, although some conotoxins were also identified in the venom bulb and the salivary gland at low levels. Obviously, these putative conotoxins from various organs were variable in diversity.

### 2.5. Diversity of Conotoxin Superfamilies

The conotoxin mixture in *C. quercinus* consisted of 133 putative conotoxins, which were classified into 34 known superfamilies, seven NSF superfamilies, and eleven unassigned superfamilies. Sequences of the A-, M-, O_1_-and I_2_-superfamilies were among the most abundant conotoxin groups in *C. quercinus* (Figure 1 and Figure 3). Despite the observed high diversity, D-, F-, G-, H-, N-, S-, and V-superfamilies were not identified in *C. quercinus*.

A-superfamily conotoxins contain six cysteine patterns (I, II, IV, VI/VII, XIV, and XXII), and they act on nicotinic acetylcholine receptors (nAChRs), the GABAB receptor and a1-adrenoceptor [33]. Here, we determined that nine conotoxins belonged to this family, with eight having the cysteine pattern I and one having the cysteine pattern XIII (Figure 4). Qc-072, Qc-073 and Qc-104 are identical to the previously identified Qc1.1b, Qc1.2 and Qc1.16 from *C. quercinus*, respectively [26,27,28]. Qc-109 is the first A-superfamily conotoxin with the cysteine framework XIII (8 cysteine residues separated by 7 loops: C-C-C-CC-C-C-C). De13.1 from *C. delessertii* with framework XIII has been characterized as a member of the G-superfamily [34]. Qc-109 shared very low sequence similarity with De13.1, but does not belong to the G-superfamily; which therefore suggests it might have exciting prospects as a functionally novel conotoxin.

M-superfamily conotoxins have eight cysteine patterns (I, II, III, IV, VI/VII, IX, XIV and XVI), and include three distinct pharmacological families, including μ-conotoxins, κM-conotoxins, and ψ-conotoxins for blocking voltage-gated sodium channels, voltage-gated potassium channels and nicotinic acetylcholine receptors, respectively [35]. However, in this study, all the related eight conotoxins presented with the cysteine pattern III (Figure 5). Qc-077 and Qc-100 were identical to Qc3.1 and Qc3-IP01 from *C. quercinus*, respectively. Qc-056 and Qc-093 were also the same as Lv3-IP02 from *C. ermineus* and Qc3-HGS02 from *C. caracteristicus*, respectively. In addition, the cysteine patterns of Qc-033 and Qc-132 were C-C.

O-superfamily members are unusually hydrophobic peptides with the same cysteine framework of C-C-CC-C-C, which provides these peptides with a similar inhibitory cystine knot (ICK) motif [36]. The O-superfamily is the most predominant superfamily in terms of conotoxin numbers (18 peptides; including O_1_, O_2_ and O_3_), but only with the VI/VII and C-C patterns of cysteine frameworks (Figure 6). Eleven O_1_-superfamily sequences were identified in our present study (Figure 6A), in which nine presented the framework VI/VII and two showed the C-C pattern. Seven O_2_-superfamily conotoxins were identified (Figure 6B), in which six presented with framework VI/VII and one possessed the C-C pattern. Qc-121 is 92% identical to the contryphan-R from *C. radiatus* [37]. Signal peptide sequence of Qc-126 indicated that this conotoxin was related to the O_3_-superfamily (Figure 6C). Interestingly, only one O_3_-superfamily was identified as sharing the cysteine framework VI/VII from *C. quercinus* [38]. Qc-126 showed 72% identity and 100% similarity to the LiCr173 from *C. lividus* [39].

I_2_-superfamily peptides have three cysteine frameworks in the primary sequences (XI, XII, and XIV), with the characteristic pattern of C-C-CC-CC-C-C. They are longer than those belonging to other *Conus* peptide families [40]. Eight I_2_-superfamily conotoxins were identified in the present study (Figure 7A), in which five presented with the framework XI, one owned the framework XII, and two possessed the framework VIII. Interestingly, framework VIII is often present in the B- and S-superfamilies, but Qc-082 and Qc-083 with the framework VIII were not similar to those conotoxins of the B- and S-superfamilies [41,42]. In fact, Qc-082 and Qc-083 are the first I_2_-superfamily conotoxins with the type VIII cysteine framework C-C-C-C-C-C-C-C-C-C. Meanwhile, six T-superfamily conotoxins were also identified (Figure 7B), in which three presented with the framework V (CC-CC), one owned the framework I (CC-C-C), and two possessed the framework XVI. For example, Qc-105 and Qc-118 showed 100% identity with Qc5.3 and Qc5.2 from *C. quercinus*, respectively.

In this study, besides these major superfamilies of conotoxins, many others of B_2_-, C-, E-, P-, J-, L-, W-, Y-, Z-superfamilies and conantokin family were also discovered in our *C. quercinus* transcriptomes. For instance, one B_2_-superfamily sequence (Qc-088) with non-cysteine, a C-superfamily sequence (Qc-101) with the framework C-C, a W-superfamily sequence (Qc-112) with non-cysteine, and one J-superfamily sequence (Qc-002) with a framework of CC-C were identified in the present work. Two P-superfamily sequences with the framework IX (Qc-124 and Qc-128) were identified in the venom gland. However, to date only nine P-superfamily conotoxins have been reported previously, and they all owned the framework IX (C-C-C-C-C-C) [43]. Meanwhile, three E-superfamily sequences with the framework XXII (Qc-024, Qc-075 and Qc-076), two Y-superfamily sequences with the framework CC-C and XVII respectively (Qc-032 and Qc-129), two Z-superfamily sequences with the framework C-C (Qc-037 and Qc-111), and two L-superfamily sequences with the framework XIV (Qc-014 andQc-123) were also identified in our current study.

Within the new superfamilies (NSF), seven conotoxins showing divergent signal regions were assigned to seven new conotoxin groups (Figure 8A). However, considering the minimal similarity of these sequences in the signal regions, we are not clear whether these groups should be recognized as distinct superfamilies or subgroups within a superfamily. In addition, within the unassigned superfamilies, there are eleven sequences without any known superfamily (Figure 8B).

### 2.6. Phylogeny of the Superfamily Signal Sequences

To investigate the relationships among these identified conotoxins from *C. quercinus*, we constructed a phylogenetic tree of all conotoxin superfamilies based on the aa sequences of the signal regions (Figure 9). This analysis showed that most of the conotoxins from *C. quercinus* transcriptomes were successfully allocated to the known superfamilies, such as B-, E, M-, O_2_- and T-superfamilies. For example, the I_2_-superfamily conotoxins are divided into three groups, including group I (Qc-082 and Qc-083), group II (Qc-098, Qc-079, Qc-053, Qc-080), and group III (Qc-106, Qc-061).

In addition, seven NSF superfamily conotoxins were also divided into five groups, including NSF-group I (Qc-004 and Qc-113) in the A-superfamily, NSF-group II (Qc-052) in the I_2_-superfamily, NSF-group III (Qc-007) in the O_1_-superfamily, NSF-group IV (Qc-050) and NSF-group V (Qc-059 and Qc-057). However, there are still a few superfamilies that cannot be assigned to any group.

## 3. Discussion

Next-generation sequencing technologies have recently been applied to study the venom duct transcriptomes of several *Conus* species, and 61~215 conotoxin sequences, belonging to 11~44 superfamilies have been discovered [1,2,13,14,15,16,17,18,19,20,21,22,23,24,25]. In previous work on *C. quercinus*, only 41 conotoxins within six superfamilies (Figure 1A) were derived from traditional approaches [26,27,28]. The next-generation transcriptome sequencing on the whole mRNA reservoir of *C. quercinus* has never been reported before. To investigate the diversity of conotoxins in a high throughput way, we employed the next-generation Illumina Hiseq2000 platform to sequence the venom duct, venom bulb and salivary gland transcriptomes of *C. quercinus*.

We obtained a total of 65, 55 and 52 putative conotoxin transcripts from the three datasets for VD, SG and VB, respectively. By comparing each two of the three transcriptomes, we determined that seven identical putative conotoxins were commonly identified in VD, SG and VB; 13 were common to both VD and VB, eight were shared by VB and SG, and five were common to both VD and SG. After removal of duplications from the three transcriptomes, we obtained a total of 133 unique putative conotoxin transcript sequences, which were classified into 34 known superfamilies with at least 1-aa difference in the mature regions between each other. In particular, conotoxin sequences of the A-, B_2_-, E-, I_2_, -M-, O_1_-, O_2_-, O_3_-, T-, Y- and Z-superfamilies were found in the VD, VB, and SG, and A-, I_2_-, M- and O-superfamilies were the most predominant conotoxin groups in *C. quercinus.* In summary, differential transcription of conotoxins happens in different organs.

Although a total of 133 putative conotoxin sequences were identified from *C. quercinus*, only seven of which were identical to the previously discovered 41 conotoxins. This remarkable difference in the same *Conus* species may be due to differences between *Conus* individuals from different geographical populations or at different developmental stages. In fact, the previously reported 41 conotoxins were pooled from different *C. quercinus* individuals. We previously also obtained various numbers of putative conotoxin sequences from different datasets (variable in organs or sequencing methods) of *C. betulinus* [24].

Early studies of the conotoxin-producing organization demonstrate that conotoxins are synthesized in a muscular, bulbous organ located at the end of the venom duct [44,45]. With the development of molecular biology, researchers found that the epithelial cells lining the cone snails’ venom duct are rich in mature mRNAs encoding conotoxins, and the venom bulb may provide the impetus to propel the venom toward the pharynx while hunting or defending [46,47]. Currently, many studies show that not all the *Conus* venom components are synthesized in the local venom duct and that other glands in the anterior gut may also make a minor contribution [17,24,48]. For example, Biggs et al. [48] discovered that PuSG1.1 and PuSG1.2 belonged to the A-superfamily with a cysteine pattern of CC-C-C in the SG of a worm-hunting species *C. pulicarius*. In addition, Lavergnea et al. [17] recently reported the discovery of 3303 (99.19%) novel conotoxin sequences in the VD, as well as 22 (0.66%) in the radular sac (RS) and 5 (0.15%) in the SG in another worm-hunting species, C. *episcopatus* by transcriptome sequencing. However, similar types of transcripts were not found in the salivary glands of several fish-hunting *Conus species* [48]. For instance, the results of Dutertre et al. [32] showed that 127 conotoxin sequences were recovered from the VD, while no conotoxin sequences were identified in the SG, and only three rare conotoxin transcripts were identified in the RS in the fish-hunting species, *C. geographus* through transcriptome sequencing and proteomics. We previously demonstrated the existence of a handful of conotoxins in the venom bulb of *C. betulinus* at very low transcription levels [24]. In this study, 65, 55 and 52 putative conotoxin sequences in the VD, SG and VB, respectively, were also identified in the worm-hunting species *C. quercinus*.

It appears that most of the identified conotoxins were synthesized in the venom duct at relatively high levels, while many conotoxins were also identified in the venom bulb and salivary gland with very low transcription levels. Although we identified a number of conotoxins in the venom bulb and salivary gland, their expression levels are relatively low. It is interesting to note that such low yields are not enough for predation, hence, we propose that these conotoxins may function endogenously within the *Conus* species [48]. In addition, these conotoxins may enhance the potency of the venom by producing bioactive components [48,49]. Thus, the precise physiological functions of these salivary gland transcripts still need to be explored further. Furthermore, identification of diverse and divergent conotoxins and new cysteine frameworks XIII and VIII in *C. quercinus* will provide a potentially fertile resource for the discovery of peptides that may target different subtypes of receptors, which provides valuable leads for drug development and biomedical applications.

## 4. Materials and Methods

### 4.1. Sample Collection and RNA Extraction

Three specimens of *C. quercinus* were collected in the offshore areas of Lingshui City, Hainan province, China. Species identification was performed using mitochondrial genes as we have reported previously [50]. The venom bulb, venom duct and salivary gland were dissected immediately before RNA extraction in TRIzol Reagent (Thermo Fisher Scientific, Waltham, MA, USA). The quality of the isolated RNA was assessed by an Agilent 2100 Bioanalyzer (Agilent Technologies, Palo Alto, CA, USA). Three Illumina cDNA libraries were constructed separately using the pooled RNA from the same organ of different individuals. Afterwards, the libraries were sequenced on an Illumina HiSeq4000 platform at BGI-Tech (BGI, Shenzhen, Guangdong, China) according to the manufacturer’s protocols (Illumina, San Diego, CA, USA).

### 4.2. Sequence Analysis and Assembling

The paired-end short reads produced from the sequencer were firstly filtered with SOAPnuke software [51] to discard contaminants with adapters and those reads with over 10% of non-sequenced bases or more than 50% of low-quality bases (base quality score ≤ 10). The remaining clean reads were assembled into contigs using Trinity v2.5.1 [52]. These obtained contigs were finally clustered based on sequence similarity and assembled to consensus unigenes using TGICL v2.1 [53]. To calculate gene transcription values in the assembled transcriptomes, clean reads were aligned to the de novo assemblies with the Bowtie 2 read aligner [54]. Then, the obtained alignments were presented to RNA-Seq by Expectation Maximization (RSEM) v1.2.31 to estimate transcript abundance in terms of FPKM [55].

### 4.3. Functional Annotation of Transcripts

Functional annotations of the identified transcripts were conducted by searching several public databases (with the threshold of E-value ≤ 10^−5^) including NCBI Nr and Nt, UniProtKB/Swiss-Prot [56], InterPro [57], Kyoto Encyclopedia of Genes and Genomes (KEGG) [58], Clusters of Orthologous Groups (COG) [59] and Gene Ontology (GO). Blast2GO v4.1 [60] was employed to perform GO annotation of the Nr blast results.

### 4.4. Prediction and Identification of Conotoxins

We applied homolog searches and an ab initio prediction method [61] to predict conotoxins from the three transcriptome datasets. For homologous prediction, all those previously known conotoxins were downloaded from the ConoServer database to construct a local reference dataset. We subsequently used BLASTX (with an E-value of 1 × 10^−5^) to run our assembled sequences against the local dataset. Those unigenes with the best hits in the BLASTX data were translated into aa sequences.

In addition, the ab initio prediction method using the HMM model [24] was adopted to discover new conotoxins. First, the three conotoxin datasets were grouped into different classes according to the superfamily and family classification in the ConoServer database. Sequences of each class were aligned with ClustalW [32], and the ambiguous results were checked by manual correction. A pHMM was built for conserved domain of each class using hmmbuild from the HMMER 3.0 package (http://hmmer.janelia.org), and the hmmsearch tool was then applied to scan every assembled unigene/EST for identification of conotoxins.

### 4.5. Classification of Gene Superfamilies

The predicted conotoxin transcripts were manually inspected using the ConoPrec: precursor analysis implemented in the ConoServer [61]. Those transcripts with duplication or truncated mature region sequences were removed. Signal peptides, gene superfamilies and cysteine frameworks of these predicted conotoxins were also checked for confirmation. Based on the 75% identity of the highly conserved signal peptide sequences [62], conotoxins could be assigned to the known superfamilies presented in the ConoServer [63]. If the conservation of a signal region was below the threshold value for any reported conotoxin superfamily, the conotoxin was regarded as a member of a new superfamily named after “NSF-Qc” plus Arabic numbers suffix. Those conotoxins without signal regions but still showing similarity either in the pro- or mature region, were considered as the Unknown group.

### 4.6. Alignment and Phylogenetic Analysis

Concatenated amino-acid alignments of the signal domains of all superfamilies were performed using MUSCLE v.3.8.31 [64]. Prior to the phylogenetic analysis, divergent and ambiguously aligned blocks were removed using Gblocks software. To choose the best-fitted model, JModelTest and Prottest3.2 [65] were used. The phylogenetic analysis was carried out using the maximum likelihood method with the RAxML8.1, and the trees was visualized and annotated using the tree viewer of MEGA4 [66]. Statistical supports were assessed with 1000 bootstrap pseudo-replicates.

### 4.7. Availability of Supporting Data

Datasets supporting the results of the present work are included within this article and related Appendix A. The transcriptome reads generated in this study have been deposited in China National GeneBank (CNGB) Nucleotide Sequence Archive (CNSA) with the accession number of CNP0000233 and the NCBI under PRJNA500655.

## 5. Conclusions

This study is the first to examine the diverse conotoxin transcription repertoire in different organs (including the venom duct, venom bulb and salivary gland) of the vermivorous Oak cone snail, *C. quercinus*. A total of 133 unique conotoxin sequences were identified by using a high-throughput transcriptome sequencing approach. These conotoxins could be classified into 34 known gene superfamilies, while 17 conotoxins were new or unassigned to any group. A-, O_1_-, M-, and I_2_-superfamilies were the most abundant, and the cysteine frameworks XIII and VIII were observed for the first time in the A- and I_2_-superfamilies, respectively. In addition, comparison of the conotoxins from the venom duct, the venom bulb and the salivary gland demonstrated that most of the identified conotoxins were synthesized in the venom duct, while a number of conotoxins were also identified in the venom bulb and salivary gland at low levels. Therefore, different organs have various conotoxins with high diversity. Interestingly, it is the first time that the existence of a number of conotoxins in the salivary gland has been revealed, although their exact functions are not clear. These novel conotoxins provide a potentially valuable resource for the development of new pharmaceuticals, and a pathway for the discovery of new conotoxins.

## Figures and Tables

**Figure 1 ijms-19-03901-f001:**
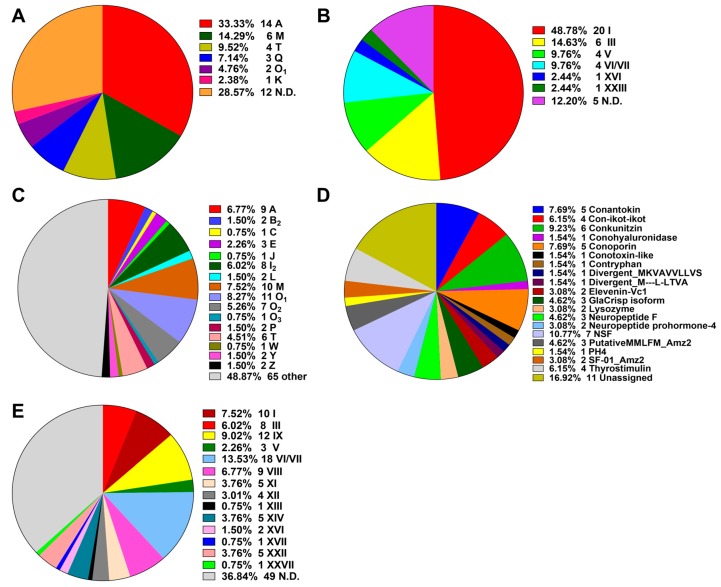
Summary of the superfamilies and cysteine patterns of conotoxins in *C. quercinus*. (**A**) The superfamilies reported previously. (**B**) The cysteine patterns reported previously. (**C**) Total superfamilies identified in this study. (**D**) Subdivision of the conotoxins summarized from the ‘Other’ group in (**C**) into further categories. (**E**) The cysteine patterns identified in this study.

**Figure 2 ijms-19-03901-f002:**
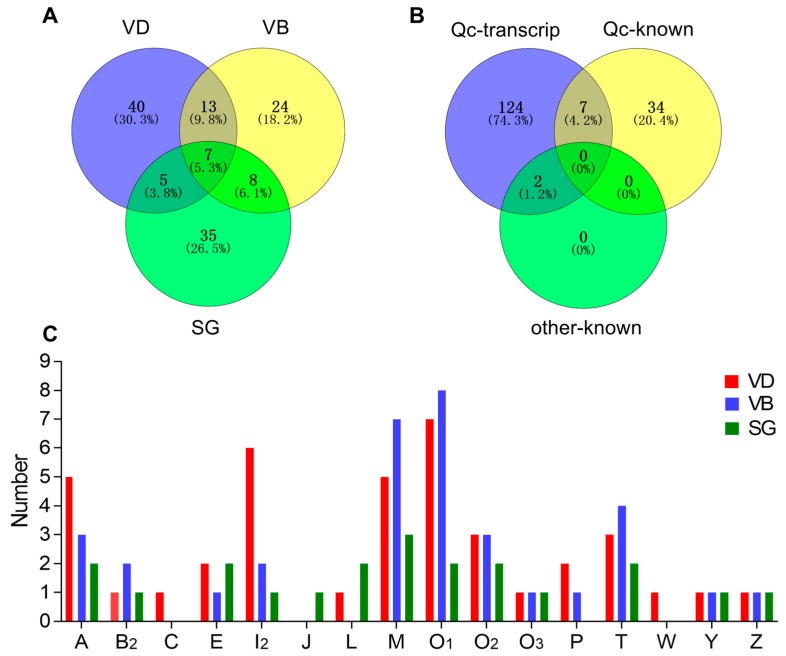
Comparison of conotoxin transcripts from various *C. quercinus* datasets. (**A**) Relationship of the identified conotoxins from the VD, VB and SG datasets. (**B**) Comparison among the total identified conotoxins from *C. quercinus* (Qc-transcript), the conotoxins reported previously in *C. quercinus* (Qc-known), and those from other cone snails (other-known). (**C**) Subdivision of 16 conotoxin superfamilies from our VD, VB and SG datasets.

**Figure 3 ijms-19-03901-f003:**
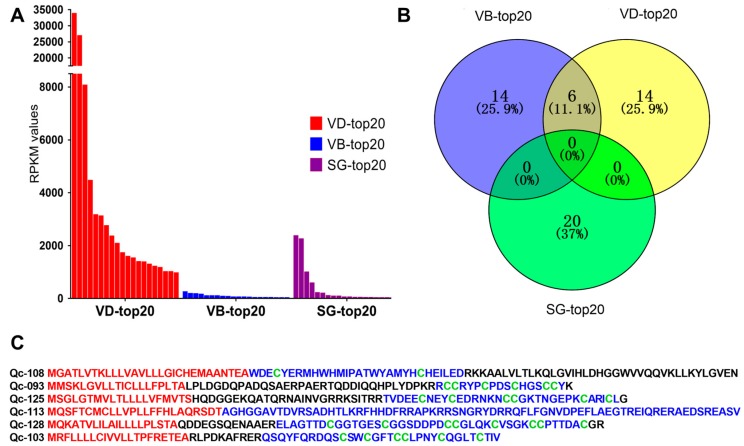
The top 20 conotoxins with the highest values of fragments per kilobase of exon per million fragments mapped (FPKM) from the three transcriptome datasets. (**A**) Comparison of the FPKM values for individual conotoxins within each dataset. (**B**) A Venn diagram of the top 20 conotoxins from each of the VD, VB and SG datasets. (**C**) Comparison of six conotoxins sequences found in the VD-top20 and VB-top20. The signal regions predicted by the ConoPrec tool are marked in red, and the mature regions (in blue) and cysteine residues (in green) are presented for comparison.

**Figure 4 ijms-19-03901-f004:**
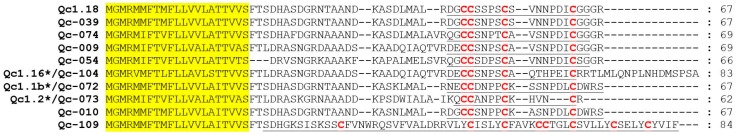
Alignment of the achieved A-superfamily sequences from our *C. quercinus* transcriptomes. * represents the conotoxins that have been reported previously. The signal regions are highlighted in yellow, the mature regions are underlined, and cysteine residues (in red) are marked for comparison.

**Figure 5 ijms-19-03901-f005:**
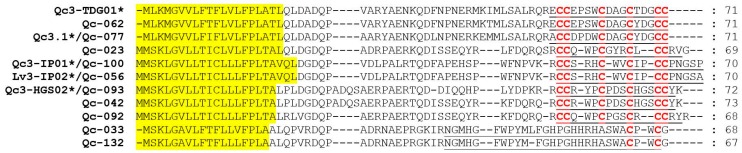
Alignment of the achieved M-superfamily sequences from our *C. quercinus* transcriptomes. * represents the conotoxins that have been reported previously. The signal regions are highlighted in yellow, the mature regions are underlined, and cysteine residues (in red) are marked for comparison.

**Figure 6 ijms-19-03901-f006:**
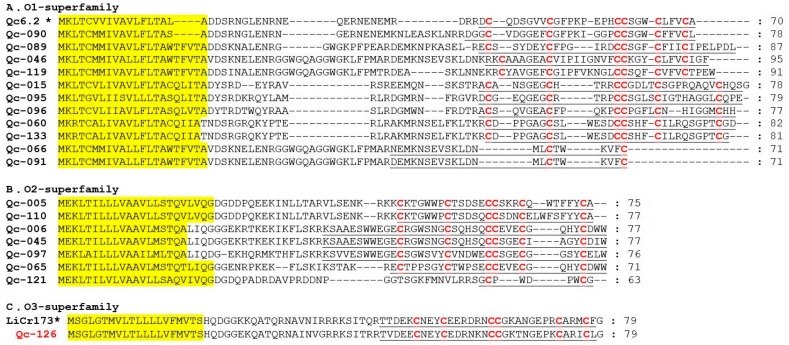
Alignment of the achieved O-superfamily sequences from our *C. quercinus* transcriptomes. (**A**) The sequences of O_1_-superfamily. (**B**) The sequences of O_2_-superfamily. (**C**) The sequences of O_3_-superfamily. * represents the conotoxins that have been reported previously. The signal regions are highlighted in yellow, the mature regions are underlined, and cysteine residues (in red) are marked for comparison.

**Figure 7 ijms-19-03901-f007:**
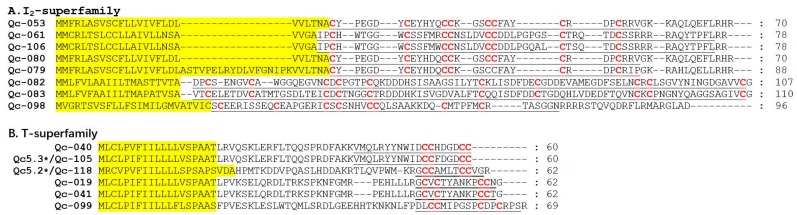
Alignment of the achieved I_2_- and T-superfamily sequences from our *C. quercinus* transcriptomes. (**A**) The sequences of I_2_- superfamily. (**B**) The sequences of T-superfamily. * represents the conotoxins that have been reported previously. The signal regions are highlighted in yellow, the mature regions are underlined, and cysteine residues (in red) are marked for comparison.

**Figure 8 ijms-19-03901-f008:**
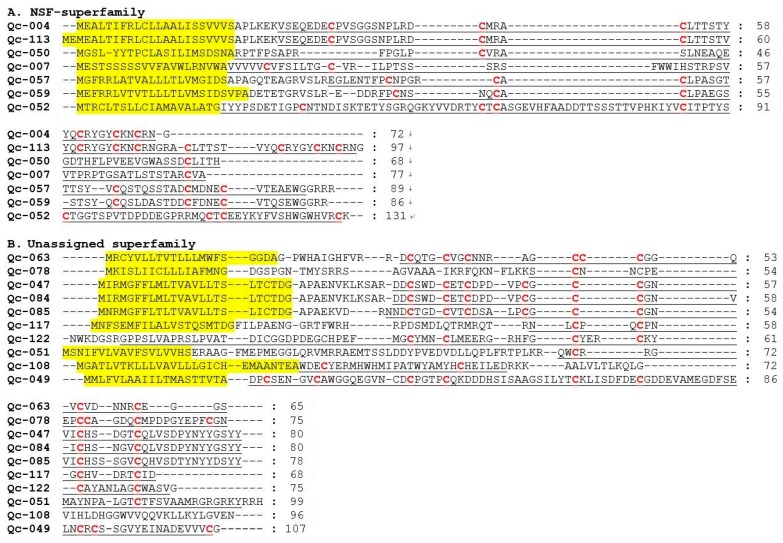
Alignment of the achieved new superfamily (NSF) and unassigned superfamily sequences from our *C. quercinus* transcriptomes. (**A**) The sequences of NSF-superfamily. (**B**) The sequences of unassigned superfamily. The signal regions are highlighted in yellow, the mature regions are underlined, and cysteine residues (in red) are marked for comparison.

**Figure 9 ijms-19-03901-f009:**
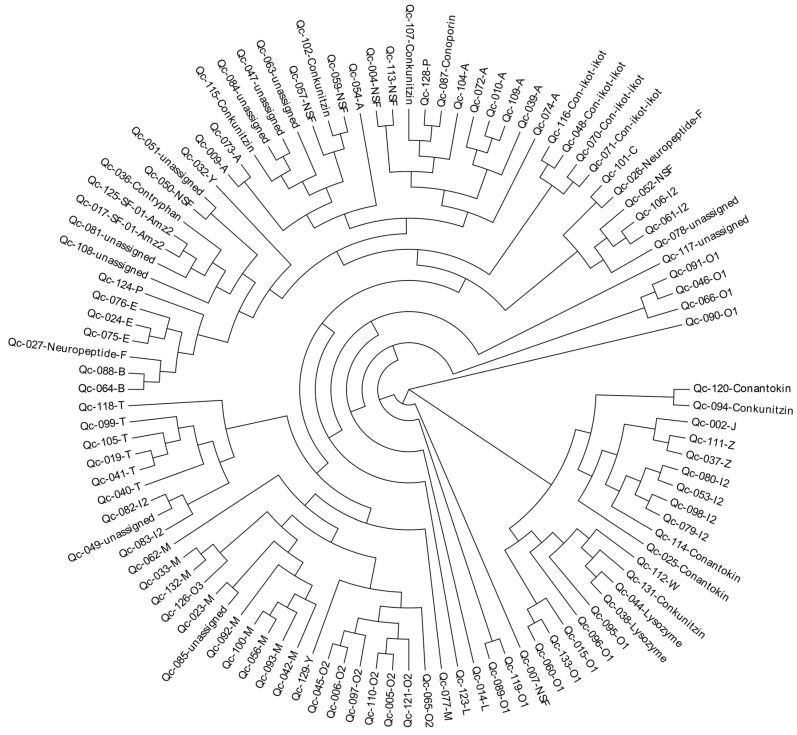
A phylogenetic tree of the signal sequences from identified conotoxins of *C. quercinus*. The phylogenetic analysis was performed using the maximum likelihood method, and the statistical supports were assessed with 1000 bootstrap pseudo-replicates.

**Table 1 ijms-19-03901-t001:** Sequencing statistics and assembly summary for the venom duct (VD), salivary gland (SG) and venom bulb (VB).

Samples	VD	SG	VB
**Raw data**			
Total Reads	29,451,202	30,184,621	30,775,070
Total length (bp)	4,417,680,300	4,527,693,150	4,616,260,500
Read length (bp)	150	150	150
**Clean data**			
Total Reads	28,865,798	28,664,492	29,290,756
Total length (bp)	4,041,211,720	4,013,028,880	4,100,705,840
Read length (bp)	140	140	140
Clean data ratio	91.48%	88.63%	88.83%
**Contigs**			
Total Number	171,606	225,404	124,936
Total Length (bp)	80,150,026	111,578,962	61,180,077
Mean Length (bp)	467	495	489
N50 (bp)	586	669	651
N70 (bp)	331	353	349
N90 (bp)	212	214	215
GC Content	43.70%	44.29%	44.32%
**Unigenes**			
Total Number	91,392	113,472	66,549
Total Length (bp)	53,668,190	72,145,553	41,085,333
Mean Length (bp)	587	635	617
N50 (bp)	800	934	884
N70 (bp)	436	485	464
N90 (bp)	254	262	259
GC Content	44.21%	44.87%	44.72%

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
