# Peer review of "High Throughput Identification of Novel Conotoxins from the Vermivorous Oak Cone Snail (*Conus quercinus*) by Transcriptome Sequencing"

_ijms, 2018, doi:10.3390/ijms19123901_

Round 1
Reviewer 1 Report
Gao and colleagues describe high-throughput transcriptome sequencing from 3 organs of the cone snail C. quercinus. The transcriptome data of venom duct, venom bulb and salivary gland showed considerable inter-organizational variations. They identified 133 putative conotoxins of which 124 were novel conotoxins. 17 of the identified contoxin sequences can be classified to unassigned conotoxin groups. A-, O1-, M-, and I2- superfamilies were the most abundant, and the cysteine frameworks XIII and VIII were observed for the first time in the A- and I2-superfamilies. This is a well written manuscript describing very interesting results. The following minor comments should be addressed:
Line 140: The authors should elucidate on the fact that only 7 of the 41 previously published C. quercinus conotoxins were identified in this study. Could this be a consequence of variation between species? This would be an interesting addition for the discussion of the manuscript which is now rather limited.
Line 163: the abbreviation FPKM should be between brackets, not the other way around.
Line 308: The observation that no conotoxin sequences could be identified from C. geographus salivary gland, while 55 putative conotoxins were identified from the salivary gland of C. quercinus, is remarkable. The authors should discuss this more in detail.
Line 326: “For the first time” contradicts with “we previously demonstrated”. This sentence is unclear and should be rewritten.
Author Response
Gao and colleagues describe high-throughput transcriptome sequencing from 3 organs of the cone snail C. quercinus. The transcriptome data of venom duct, venom bulb and salivary gland showed considerable inter-organizational variations. They identified 133 putative conotoxins of which 124 were novel conotoxins. 17 of the identified contoxin sequences can be classified to unassigned conotoxin groups. A-, O1-, M-, and I2- superfamilies were the most abundant, and the cysteine frameworks XIII and VIII were observed for the first time in the A- and I2-superfamilies. This is a well written manuscript describing very interesting results. The following minor comments should be addressed:
Line 140: The authors should elucidate on the fact that only 7 of the 41 previously published C. quercinus conotoxins were identified in this study. Could this be a consequence of variation between species? This would be an interesting addition for the discussion of the manuscript which is now rather limited.
Answer: Thanks for your nice advice. Yes, we made more discussions in the revised manuscript. Please find more details on lines 321-327.
Line 163: the abbreviation FPKM should be between brackets, not the other way around.
Answer: Yes, it is done on line 165.
Line 308: The observation that no conotoxin sequences could be identified from C. geographus salivary gland, while 55 putative conotoxins were identified from the salivary gland of C. quercinus, is remarkable. The authors should discuss this more in detail.
Answer: Yes, related discussion was provided on lines 332-346.
Line 326: “For the first time” contradicts with “we previously demonstrated”. This sentence is unclear and should be rewritten.
Answer: “For the first time” was deleted.
By the way, we checked the manuscript carefully with editing help from a colleague, who had worked in the USA for over nine years.
Reviewer 2 Report
This paper was well written, informative, and content rich. The work presented was solidly presented and robust with respect to results. There were a few areas where presentation could be improved. For instance, the listing of large numbers in series was confusing given the dual use of commas in different capacities (e.g. lines 75-80). Recommend a different representation for the sequential numbers. In line 113, change ConoServer databa to ConoServer database. In lines 123 to 130, either use a space or don't use a space when presenting the cysteine frameworks. Be consistent in the presentation of the material. The wording of the sentence in lines 272-275 needs to be modified for clarity. Lines 327-329: there is no point of reference for the statement, "relatively low." What is a low value? Be more explicit in this statement. Lines 340-341: sentence needs to be rewritten for clarity and content.
Author Response
This paper was well written, informative, and content rich. The work presented was solidly presented and robust with respect to results. There were a few areas where presentation could be improved.
For instance, the listing of large numbers in series was confusing given the dual use of commas in different capacities (e.g. lines 75-80). Recommend a different representation for the sequential numbers.
Answer: Thanks for your advice. We rewrote these numbers on lines 77, 80 and 84.
In line 113, change ConoServer databa to ConoServer database.
Answer: Yes, it is done on line 114.
In lines 123 to 130, either use a space or don't use a space when presenting the cysteine frameworks. Be consistent in the presentation of the material.
Answer: Thanks for your advice. According to your suggestion, we added a space in all the cysteine frameworks throughout the revised manuscript.
The wording of the sentence in lines 272-275 needs to be modified for clarity.
Answer: We rephrased these sentences on lines 277-281.
Lines 327-329: there is no point of reference for the statement, "relatively low." What is a low value? Be more explicit in this statement.
Answer: We tried to describe the comparative transcription levels among the three transcriptomes. In fact, the highest RPKM values of all conotoxins in the SG-top20 and VB-top20 datasets were 2,364.27 and 247.12, respectively. However, the transcription levels of the most peptides in SG were less than 200. That is why we mentioned “relatively low”. We deleted this sentence in the revised manuscript.
Lines 340-341: sentence needs to be rewritten for clarity and content.
Answer: Yes, it is done on lines 369-373.
By the way, we checked the manuscript carefully with editing help from a colleague, who had worked in the USA for over nine years.
Reviewer 3 Report
The authors of High Throughput Identification of Novel Conotoxins from Vermivorous Oak Cone Snail (Conus quercinnus) by Transcriptome Sequencing have done a thorough transcriptome analysis of the venom gland, venom duct and salivary gland. This is the most exhaustive transcriptome study to date and reveals a number of new superfamilies and several new conotoxins carrying different frameworks. The results are interesting, but seems a bit convoluted and would benefit from some rearrangement to increase the flow of the text. However, the work is thorough and would be of interest to readers of IJMS in general and specifically the conopeptide community. Please see comments on the manuscript below.
Abstract:
Line 35-36 Please rephrase the concluding sentence. It is a bit confusing, especially how investigating various organs will create a new pathway for further high-throughput discovery.
Introduction:
Line 52 the sentence containing “unmanageable chronic pains in serious cancer” would benefit from being reworded.
Line 56 “Lowly sensitive” should be reworded as it isn’t grammatically correct
Line 53-63 What is meant by traditional approaches is unclear. What traditional approaches are they talking about and how does this study’s methods differ? Do they mean assay-guided fractionations or proteomics or less sensitive transcriptomics methodology being used in the past? How is the work in this study an improvement?
Line 57 Please explain “higher sequencing depth”
Results:
Figure 1. The pie charts are very busy and the colours are difficult to decipher. Also, the font of the legends is too small to be read.
Figure 2. Please change the font to Arial or similar in the Venn diagrams in A and B. Also, the colours in panel C are very difficult to decipher and the X-axis legend in panel C shouldn’t be on an angle.
Figure 3. It is unclear why the Cys-frameworks aren’t aligned? Also the font should be fixed in panel B. See above.
Figure 4, 5, 6, 7, 8. The font is very small and it is very busy with all the colours.
Line 231. If something is 100% identical, there can’t be any differences between them. Please rephrase the sentence.
Line 235 unsure what is meant by “owned the C-C pattern”, also in line 254
Line 228. Cys residues involved in disulfide bonds are called cystines (not cysteines). Please fix the inhibitory cysteine know (ICK) motif.
In summary section 2.2 and 2.5 there appears to be quite a bit of repetition and the writing gets a bit impenetrable. Perhaps a better way to organize it would be one section summarizing what was found from the venom duct, the next section summarizing the venom bulb, the next section summarizing the salivary gland. Then the next section could compare and contrast the three organs and include some bar graphs and Venn Diagrams. Including an improved section of the rationale behind the new superfamily classification would also improve the manuscript.
Discussion:
Line 321 is the I2 superfamily different from the I2 superfamily?
Line 327 The sentence states “For the first time, as previously demonstrated”. Both can’t be true.
The discussion could be vastly improved by a further analysis of the results and not just a resummary of the results section. There is a lot more that can be derived from the data presented in this exciting study. This may include discussing what does the different levels of expression of conotoxins mean about the origins of the salivary glands as opposed to the venom glands? Perhaps one of the salivary glands adapted to perform a new function? Do their results support that? Perhaps the results do not support that; would be even more interesting. The discussion mentions the radular sac, why didn’t this study examine the radular sac too? Too small? Not enough samples? What about all those new superfamilies they found? More on that would be great.
Conclusion:
Line 341 the sentence “conotoxin from different organs have been rich in diversity” has been repeated several times throughout the manuscript and is grammatically incorrect. Please reword it.
Methods and Materials:
How many snails did the authors use? Did they pool the RNA?
What reagent did they use to extract the RNA? (TRIZOL, RNAeasy, etc.)?
Line 364 e < 10-5 is really really adequate for assigning functionality. e<10-10 would have been better with e<10-20 is being preferable since it is the most rigorous. Was it simply the low threshold that changed their results? Could this be why they found so many new superfamilies? What would happen if they were more stringent?
Line 394 Could removing divergent and ambiguously aligned blocks before phylogenetic analysis influence the phylogenetic analysis??
Author Response
The authors of High Throughput Identification of Novel Conotoxins from Vermivorous Oak Cone Snail (Conus quercinnus) by Transcriptome Sequencing have done a thorough transcriptome analysis of the venom gland, venom duct and salivary gland. This is the most exhaustive transcriptome study to date and reveals a number of new superfamilies and several new conotoxins carrying different frameworks. The results are interesting, but seems a bit convoluted and would benefit from some rearrangement to increase the flow of the text. However, the work is thorough and would be of interest to readers of IJMS in general and specifically the conopeptide community. Please see comments on the manuscript below.
Abstract
Line 35-36 Please rephrase the concluding sentence. It is a bit confusing, especially how investigating various organs will create a new pathway for further high-throughput discovery.
Answer: Thanks for your food advice. We rewrote this sentence as follows (on lines 34-36).
Therefore, various organs have different conotoxins with high diversity, suggesting more contributions from more organs to the high-throughput discovery of new conotoxins for future drug development.
Introduction
Line 52 the sentence containing “unmanageable chronic pains in serious cancer” would benefit from being reworded.
Answer: We rewrote it as “has been approved by the American FDA for treatment of chronic pains in patients with cancer or AIDS.” Please find more details on lines 51-52.
Line 56 “Lowly sensitive” should be reworded as it isn’t grammatically correct
Answer: It was changed to “low sensitivity” on line 56.
Line 53-63 What is meant by traditional approaches is unclear. What traditional approaches are they talking about and how does this study’s methods differ? Do they mean assay-guided fractionations or proteomics or less sensitive transcriptomics methodology being used in the past? How is the work in this study an improvement?
Answer: Sorry for the misleading descriptions. In fact, traditional methods refer to the traditional methods for separation and purification of venom proteins. We therefore changed it to “these methods to isolate and identify the potential bioactive substances directly from venoms are now widely considered to be time-consuming, low sensitivity and often limited by sample availability.” Please find more details on lines 54-56.
Line 57 Please explain “higher sequencing depth”
Answer: Transcriptome sequencing only sequences the transcribed coding regions, which have been amplified for more sequencing reads. That is to say, higher sequencing depth is realized in this high-throughput sequencing technique. We added “(amplification with more sequencing reads)” on lines 57-58.
Results:
Figure 1. The pie charts are very busy and the colours are difficult to decipher. Also, the font of the legends is too small to be read.
Answer: Figure 1 was revised according to your suggestion. Please find more details in the new Figure 1 on page 4.
Figure 2. Please change the font to Arial or similar in the Venn diagrams in A and B. Also, the colours in panel C are very difficult to decipher and the X-axis legend in panel C shouldn’t be on an angle.
Answer: Yes, it is done. Please find more details in the revised Figure 2 on page 5.
Figure 3. It is unclear why the Cys-frameworks aren’t aligned? Also the font should be fixed in panel B. See above.
Answer: Figure 3 was revised according to your kind suggestion. In fact, the Cys-frameworks of the six peptides are in different types. It is therefore difficult to align them. However, we highlighted the cysteine residues in green for your convenient comparison.
Figure 4, 5, 6, 7, 8. The font is very small and it is very busy with all the colours.
Answer: Sorry for this inconvenience. We revised these figures (on pages 7-10) according to your suggestion.
Line 231. If something is 100% identical, there can’t be any differences between them. Please rephrase the sentence
Answer: This sentence was deleted in the revised manuscript.
Line 235 unsure what is meant by “owned the C-C pattern”, also in line 254
Answer: The C-C pattern is a special cysteine framework, such as xxCxCxx.
Line 228. Cys residues involved in disulfide bonds are called cystines (not cysteines). Please fix the inhibitory cysteine know (ICK) motif.
Answer: Yes, it is corrected on line 234 as “inhibitory cystine knot (ICK)”.
In summary section 2.2 and 2.5 there appears to be quite a bit of repetition and the writing gets a bit impenetrable. Perhaps a better way to organize it would be one section summarizing what was found from the venom duct, the next section summarizing the venom bulb, the next section summarizing the salivary gland. Then the next section could compare and contrast the three organs and include some bar graphs and Venn Diagrams. Including an improved section of the rationale behind the new superfamily classification would also improve the manuscript.
Answer: Thanks for your suggestion. We would rather keep the present organization, since it would be easier for us to compare the differences among the three transcriptomes. However, we added more details in both sections for more clarity.
Discussion:
Line 321 is the I2 superfamily different from the I2 superfamily?
Answer: Sorry for this mistake. We changed it throughout the revised manuscript.
Line 327 The sentence states “For the first time, as previously demonstrated”. Both can’t be true.
Answer: “For the first time” was deleted in the revised manuscript.
The discussion could be vastly improved by a further analysis of the results and not just a resummary of the results section. There is a lot more that can be derived from the data presented in this exciting study. This may include discussing what does the different levels of expression of conotoxins mean about the origins of the salivary glands as opposed to the venom glands? Perhaps one of the salivary glands adapted to perform a new function? Do their results support that? Perhaps the results do not support that; would be even more interesting. The discussion mentions the radular sac, why didn’t this study examine the radular sac too? Too small? Not enough samples? What about all those new superfamilies they found? More on that would be great.
Answer: Yes, we added more discussions under this section. Please find more details on lines 332-346. For the radular sac, however, we didn’t collect any samples since it is not our target. We will examine it in the next project.
Conclusion:
Line 341 the sentence “conotoxin from different organs have been rich in diversity” has been repeated several times throughout the manuscript and is grammatically incorrect. Please reword it.
Answer: Thanks for your advice. This sentence was revised as follows on lines 369-370.
Therefore, different organs have various conotoxins with high diversity.
It was also changed in the Abstract (line 34) and Results (lines 161, 190, 370).
Methods and Materials:
How many snails did the authors use? Did they pool the RNA?
Answer: We provided more details on lines 376 and 381. In fact, three snails were used in the study, and mRNA of the same tissue from three snails was pooled for sequencing.
What reagent did they use to extract the RNA? (TRIZOL, RNAeasy, etc.)?
Answer: RNA was extracted using TRIzol. We added more details on lines 379-380.
Line 364 e < 10-5 is really really adequate for assigning functionality. e<10-10 would have been better with e<10-20 is being preferable since it is the most rigorous. Was it simply the low threshold that changed their results? Could this be why they found so many new superfamilies? What would happen if they were more stringent?
Answer: In fact, e < 10-5 is the most commonly used threshold in BLAST and alignment researches. If we use a more stringent threshold, we will definitely obtain highly-similar but less conotoxins and superfamilies; however, we may have missed those non-conserved ones and the new superfamilies. We usually feel reliable using a threshold of e < 10-5 for most studies.
Line 394 Could removing divergent and ambiguously aligned blocks before phylogenetic analysis influence the phylogenetic analysis??
Answer: This is a very good suggestion. According to your suggestion, we revised Figure 9 on page 11.
By the way, we checked the manuscript carefully with editing help from a colleague, who had worked in the USA for over nine years.
Reviewer 4 Report
The manuscript of Bingmiao Gao et al. describes the results obtained in a transcriptomic study aimed at realizing large-scale discovery of conotoxin sequences from different organs of the vermivorous Conus quercinus, using high-throughput transcriptome sequencing.
The study is based on robust and innovative methodologies in the frame of the "omics" studies that may increase the soundness of the research. The conclusions are fairly supported by the results and may be usefull to contribute to the cataloguing of conotoxin diversity in the main lineages of cone snails.
Basing on these observation, I believe that the manuscript may be suitable for its publication in IJMS, after addressing the following minor points that may increase its quality:
- In the conclusion section and in the last part of discussion, authors may discuss and speculate the possible outcome of this research and the exploitment of these results for further studies.
- In the material and methods section, authors should specify the number of Conus quercinus individuals used for the research. How they classified the animal as Conus quercinus? Was taxonomic identification carried out?
Author Response
The manuscript of Bingmiao Gao et al. describes the results obtained in a transcriptomic study aimed at realizing large-scale discovery of conotoxin sequences from different organs of the vermivorous Conus quercinus, using high-throughput transcriptome sequencing.
The study is based on robust and innovative methodologies in the frame of the "omics" studies that may increase the soundness of the research. The conclusions are fairly supported by the results and may be usefull to contribute to the cataloguing of conotoxin diversity in the main lineages of cone snails.
Basing on these observation, I believe that the manuscript may be suitable for its publication in IJMS, after addressing the following minor points that may increase its quality:
- In the conclusion section and in the last part of discussion, authors may discuss and speculate the possible outcome of this research and the exploitment of these results for further studies.
Answer: Thanks for your advice. We added a few more sentences in the Discussion and Conclusion sections to address these questions. Please find more details on lines 371-373 and 353-358.
- In the material and methods section, authors should specify the number of Conus quercinus individuals used for the research. How they classified the animal as Conus quercinus? Was taxonomic identification carried out?
Answer: Thanks for your good questions and advice. The number of C. quercinus was provided on line 376. For the classification with several mitochondrial genes, please find more details in our previous report (Gao et al., 2018, PLos One, 13: e0193053) and lines 377-378 in the revised manuscript.
By the way, we checked the manuscript carefully with editing help from a colleague, who had worked in the USA for over nine years.